

# Using demographic data to understand the distribution of H1N1 and COVID-19 pandemics cases among federal entities and municipalities of Mexico

Yohanna Sarria-Guzmán[1,2], Jaime Bernal[3], Michele De Biase[4], Ligia C. Muñoz-Arenas[5], Francisco Erik González-Jiménez[6], Clemente Mosso[1], Arit De León-Lorenzana[7] and Carmine Fusaro[8]

[1] Centro Regional de Investigación en Salud Pública, Instituto Nacional de Salud Pública, Tapachula, Chiapas, Mexico
[2] Facultad de Ingeniería y Ciencias Básicas, Fundación Universitaria del Área Andina, Valledupar, Cesar, Colombia
[3] Facultad de Medicina, Universidad del Sinú, Cartagena de Indias, Bolivar, Colombia
[4] Dipartimento di Ingegneria Ambientale, Università della Calabria, Rende, Calabria, Italy
[5] Facultad de Ingeniería Ambiental, Universidad Popular Autónoma del Estado de Puebla, Puebla, Puebla, Mexico
[6] Facultad de Ciencias Químicas, Universidad Veracruzana, Orizaba, Veracruz, Mexico
[7] Instituto de Ecología, Universidad Nacional Autónoma de México, Merida, Yucatan, Mexico
[8] Facultad de Ingenierías, Universidad de San Buenaventura—Cartagena, Cartagena de Indias, Bolivar, Colombia

Corresponding author
Carmine Fusaro, cafu18685@gmail.com

## ABSTRACT

**Background:** The novel coronavirus disease (COVID-19) pandemic is the second global health emergency the world has faced in less than two decades, after the H1N1 Influenza pandemic in 2009–2010. Spread of pandemics is frequently associated with increased population size and population density. The geographical scales (national, regional or local scale) are key elements in determining the correlation between demographic factors and the spread of outbreaks. The aims of this study were: (a) to collect the Mexican data related to the two pandemics; (b) to create thematic maps using federal and municipal geographic scales; (c) to investigate the correlations between the pandemics indicators (numbers of contagious and deaths) and demographic patterns (population size and density).

**Methods:** The demographic patterns of all Mexican Federal Entities and all municipalities were taken from the database of "Instituto Nacional de Estadística y Geografía" (INEGI). The data of "Centro Nacional de Programas Preventivos y Control de Enfermedades" (CENAPRECE) and the geoportal of Mexico Government were also used in our analysis. The results are presented by means of tables, graphs and thematic maps. A Spearman correlation was used to assess the associations between the pandemics indicators and the demographic patterns. Correlations with a $p$ value < 0.05 were considered significant.

**Results:** The confirmed cases (ccH1N1) and deaths (dH1N1) registered during the H1N1 Influenza pandemic were 72.4 thousand and 1.2 thousand respectively. Mexico City (CDMX) was the most affected area by the pandemic with 8,502 ccH1N1 and 152 dH1N1. The ccH1N1 and dH1N1 were positively correlated to demographic patterns; $p$-values higher than the level of marginal significance were

found analyzing the % ccH1N1 and the % dH1N1 vs the population density.
The COVID-19 pandemic data indicated 75.0 million confirmed cases (ccCOVID-19) and 1.6 million deaths (dCOVID-19) worldwide, as of date. The CDMX, where 264,330 infections were recorded, is the national epicenter of the pandemic.
The federal scale did not allow to observe the correlation between demographic data and pandemic indicators; hence the next step was to choose a more detailed geographical scale (municipal basis). The ccCOVID-19 and dCOVID-19 (municipal basis) were highly correlated with demographic patterns; also the % ccCOVID-19 and % dCOVID-19 were moderately correlated with demographic patterns.
**Conclusion:** The magnitude of COVID-19 pandemic is much greater than the H1N1 Influenza pandemic. The CDMX was the national epicenter in both pandemics. The federal scale did not allow to evaluate the correlation between exanimated demographic variables and the spread of infections, but the municipal basis allowed the identification of local variations and "red zones" such as the delegation of Iztapalapa and Gustavo A. Madero in CDMX.

## INTRODUCTION

Disease outbreaks have frequently occurred in human history and have had significant impacts on many sectors such as public health, political and economic systems at a local and global scale. The novel coronavirus disease (COVID-19) pandemic is the second global health emergency that governments, health organizations and citizens worldwide have faced over the last two decades (*WHO, 2020*).

The widely known as "Swine flu", the first influenza pandemic of the 21 century, occurred in 2009–2010 (*CDC, 2020*) caused by the virus A (H1N1) pdm09; the novel virus presented genetic material from human, pig and bird flu viruses. It was very different from H1N1 viruses formerly identified and, therefore, vaccination of seasonal flu offered little cross-protection (*Fraser et al., 2009*; *Lim & Mahmood, 2011*). Most young people did not show immunity to this virus (*Crum-Cianflone et al., 2009*), but around one-third of the elderly manifested antibodies probably acquired from exposure to other H1N1 virus in their youth (*CDC, 2020*). The symptoms of the Influenza A (H1N1) 2009 in people have ranged from mild to severe and include fever, cough, sore throat, nose, body aches, headache, chills, fatigue, diarrhea and vomiting (*Witkop et al., 2010*; *Petrova & Russell, 2018*; *Saunders-Hastings et al., 2017*). There were approx. 2.8 million Influenza cases and between 151.7 thousand to 575.4 thousand deaths worldwide (April 2009–April 2010) mostly people younger than 65 years (*CDC, 2020*). Many countries, especially in Africa and Southeast Asia, lacked the ability to perform detection tests and still today, there is no certainty about factual pandemic deaths (*WHO, 2020*). Mexico was among the first countries to notify the World Health Organization (WHO) of viral respiratory disease "Swine flu" and was, also, one of the most affected nations by this pandemic

(*Kain & Fowler, 2019*; *CDC, 2020*). The H1N1 Influenza confirmed cases (ccH1N1) were approx. 72.4 thousand while the deaths (dH1N1) were 1.2 thousand (data updated to 10th April, 2010) (http://www.cenaprece.salud.gob.mx/). On August 10th of 2010, the WHO declared the end of sanitary emergency related to Influenza A (H1N1) 2009 pandemic. However, the A (H1N1) pdm09 virus continues to circulate as a seasonal flu virus causing several hospitalizations and deaths in many developing countries every year (*WHO, 2020*).

At the end of 2019, cases of pneumonia associated with the novel coronavirus (SARS-CoV-2) were reported in the Chinese province of Wuhan. The SARS-CoV-2 is the seventh member of the coronavirus family to infect humans. It is characterized by prolonged incubation time and shedding from asymptomatic patients (*Widders, Broom & Broom, 2020*). The COVID-19 outbreak spread quickly worldwide avoiding medical control and it was declared pandemic by the WHO on the 11th of March 2020 (*WHO, 2020*). The SARS-CoV-2 infected patients are mostly asymptomatic or experience mild symptoms such as fever, dry cough, and sore throat (*Ma et al., 2020*). However, some patients, principally advanced age people or individuals with comorbities, develop severe and even fatal medical complications (*Mehta et al., 2020*; *Sohrabi et al., 2020*).

Risk factors and health conditions that could aggravate the clinical picture of patients with COVID-19 include hypertension, diabetes, cardiovascular disease, chronic respiratory disease, chronic kidney disease, immune compromised status, neurological disorders, cancer, smoking and obesity (*Alqahtani et al., 2020*; *Nieman & Wentz, 2019*; *Roncon et al., 2020*). People with diabetes increased the odds to develop severe symptoms or die from COVID-19 by 2.7 (*Williamson et al., 2020*). Patients that suffer from hypertension, cardiovascular or cerebrovascular disease were up to three times more likely to have severe COVID-19 symptoms than people in good health (*Wang et al., 2020*). *Williamson et al. (2020)* suggested that the risk of mortality from COVID-19 increase in patients with asthma or respiratory disease. The risk of severe symptoms or death from COVID-19 significantly increase in people with cancer, moreover, among patients with blood cancers the mortality rate is twice that of patients with solid tumors (*Meng et al., 2020*). According to *Simonnet et al. (2020)*, people that suffer from obesity increase seven times the odds of developing severe COVID-19.

Environmental issues such as air pollution could compromise lung function, and consequently increase the vulnerability to respiratory infections such as COVID-19. It has been proved that other local factors such us overcrowded areas, frequent social interactions and full public transport can increase the number of contagious by SARS-CoV-2 due its mechanism of transmission; the virus can persist in aerosols and surfaces. With this in mind, the proximity and contact among citizens result key determinants (*Meyerowitz et al., 2021*; *Jordan, Adab & Cheng, 2020*).

The incidence and morbidity of COVID-19 were different among countries and even regions of the same country, probably due to the different implementation and timing of mitigation strategies adopted by governments (social distancing, travel restrictions and quarantine) (*Gerli et al., 2020*) and social demographic indicators (population density, poverty and unemployment) (*Ramírez & Lee, 2020*). The SARS-CoV-2 has infected more than 75.0 million people and we have approx. 1.6 million deaths worldwide by COVID-19

as of 18th December, 2020 (*Coronavirus Research Center, 2020*). The WHO indicates Latin America as "red zone" of coronavirus transmissions in the world (3rd June, 2020). More specifically, in Mexico the ccCOVID-19 (1.2 million) and dCOVID-19 (115.1 thousand) are inexorably increasing day by day (data updated to 16th December, 2020) (*COVID-19 México, 2020*).

Spread of infectious diseases is frequently associated with increase in population size, population density, aged population density and construction land area (*Vazquez-Prokopec et al., 2010*; *Barreto et al., 2011*; *Fang et al., 2012*; *Pequeno et al., 2020*; *Poole, 2020*; *Rodrigue, Luke & Osterholm, 2020*; *Verity et al., 2020*). Epidemics in smaller cities are generally focused on a shorter period (*Dalziel et al., 2018*), while the incidence is diffuse in the bigger urban areas leading to more interaction among local residents, workers and tourist which make them potential hotspots for pandemics spread; therefore, communication routes such as national highways or freeways, and public transport that is, subway, buses and trams could magnify the impact of the outbreaks. Understanding the spatial distribution of the COVID-19 confirmed cases (ccCOVID-19), COVID-19 deaths (dCOVID-19) and their association with demographic factors can benefit to isolate the areas of highest risk of infection and the most vulnerable groups of the population at local scale. The geographical scales (national, regional or local scale) are key elements in determining the correlations between demographic factors and spread of outbreaks.

The aims of this article were: (a) to collect the official Mexican data (counts of confirmed cases and deaths) related to the H1N1 Influenza pandemic (*CENAPRECE, 2020*) and the official governmental data related to the COVID-19 pandemic (*COVID-19 México, 2020*) both on a federal and municipal scale; (b) to produce thematic maps (population size—confirmed cases, population size—deaths); (c) to investigate the correlations between the pandemics indicators and demographic patterns.

## MATERIALS AND METHODS

### Country context

Mexico is constituted by 32 Federal Entities (FEs) and covers a geographical area of 1,964,375 km$^2$. It is the second most populated country in Latin America after Brazil. Its population, that has increased in the last 10 years (+13.72%), passing from 112.34 to 127.79 million inhabitants, is mainly grouped in urban areas. All of FEs have reported an upward trend, although the increase in population has been slower than in the first decade of this century.

The population density (2020) varied significantly among the FEs and ranged between 11 inhabitants per km$^2$ (hab./km$^2$) in Baja California Sur and 6,033 hab./km$^2$ in Mexico City (CDMX). The capital is far more crowded than any other city in Mexico and is one of the most populated cities in North America, with 9.02 million inhabitants distributed in the "relatively small" surface area of 1,495 km$^2$. Its population density is higher than the national average (65 hab./km$^2$); in particular, the delegations of Iztapalapa (1.82 million hab.) and Gustavo A. Madero (1.18 million hab.) are among the most overpopulated areas in the world. The state of Mexico (22,351 km$^2$), in the central part of

the country, neighboring with CDMX, is the most populated among the FEs (17.43 million hab.–780 hab./km$^2$), its inhabitants are concentrated mainly in the cities of Ecatepec de Morelos (1.71 million hab.) and Nezahualcóyotl (1.13 million hab.). Tlaxcala, although is the second smallest Mexican State in size (4,016 km$^2$), is in the fourth position in the ranking of the most densely populated places of the republic (344 hab./km$^2$). Its residents live mostly in Tlaxcala city (103,435 hab.). Guerrero (63,596 km$^2$–3.66 million hab.) in the Pacific coast and Yucatan (39,524 km$^2$–2.26 million hab.) in the Gulf of Mexico reported a population density concordant with the national average. Acapulco de Juárez (840,795 hab.) and Merida (963,861 hab.) are the urban areas more populated in addition to being among the significant national and international tourist destinations. Baja California Sur (73,909 km$^2$–804,708 hab.) in the Pacific coast and Durango (123,317 km$^2$–1.87 million hab.) in the central part of the country are among the least populated FEs, with population density about five times lower than the national average.

The demographic patterns that is, population size (hab.), population density (hab./km$^2$) and surface (km$^2$) characteristics of each one of thirty-two FEs and all Mexican municipalities were taken from the database of "Instituto Nacional de Estadística y Geografía" of Mexico (INEGI, 2020). The topographic map (scale 1:250,000) updated to 2018 was adopted as cartographic base (CONABIO, 2020).

## Pandemics data

As stated before, risk factors that could aggravate the symptoms derived by COVID-19 include hypertension, diabetes, chronic respiratory or cardiovascular disease, immune compromised status, neurological disorders, cancer, smoking and obesity (Alqahtani et al., 2020; Nieman & Wentz, 2019; Roncon et al., 2020). In addition, various factors such overcrowded areas, frequent social interactions, full public transport, poor attention to preventive measures in particular disrespect the social distancing could be involved in the growth of the number of contagious during pandemics, which cause the collapse of the national health system and the drastic reduction in the number of ICUs available in Mexican hospitals.

Using the official Mexican data, we have chosen to analyze only the correlation between demographic patterns that is, population size and density with the number of contagious and victims in the two pandemics (Pequeno et al., 2020).

The official data of CENAPRECE (http://www.cenaprece.salud.gob.mx/) related to ccH1N1 and dH1N1 were used in this analysis while the geoportal of Mexico's government (COVID-19 México, 2020) was adopted to find the data of ccCOVID-19 and deaths dCOVID-19 related to the pandemic.

The results are presented as tables, graphs and thematic maps. The thematic maps were obtained using ArcMap 10.5 (ESRI, Redlands, CA, USA); two attributes have been drawn (population size—counts of confirmed cases or counts of deaths) in each map. A special focus was directed to CDMX.

A Spearman correlation was performed to assess the associations between the pandemics data and demographic patterns. Correlations with $p$ values < 0.05 were considered significant. The Spearman coefficient values range between −1 and 1and

indicate negative or positive correlation, the values close to zero indicate poor or no correlation between the variables.

The nonparametric method locally weighted linear regression (LOESS), developed thought the programing language Python, have been used according to *Austin & Steyerberg (2014)* for the estimation of relationships between the demographic and pandemics indicators; the results were presented through XY graphs.

# RESULTS

## The H1N1 influenza pandemic

The first ccH1N1 were recorded in the small rural town of La Gloria, Veracruz (central part of Mexico) at the end of March, 2009. The national ccH1N1 and dH1N1 registered by CENAPRECE (http://www.cenaprece.salud.gob.mx/) during the sanitary emergency were 72.4 thousand and 1.2 thousand respectively.

The thematic maps (Figs. 1A and 1B) describe the spatial distribution of ccH1N1 and dH1N1 in Mexico and their association with population density. The CDMX was the most affected area by the "Swine flu" pandemic with 8,502 ccH1N1 and 152 dH1N1. Other four FEs passed the threshold of 4,000 ccH1N1 that is, State of Mexico (4,682 ccH1N1), San Luis Potosí (4,446 ccH1N1), Nuevo León (4,358 ccH1N1) and Jalisco (4,333 ccN1N1). The ccH1N1 were less than a thousand people in six FEs that is, Baja California Sur, Morelos, Quintana Roo, Sinaloa, Coahuila and Campeche. The ccH1N1 ranged between 1,000 and 4,000 in the rest of the FEs (Table 1).

The Spearman correlation at federal scale (Fig. 2A) indicated different results. The ccH1N1 and dH1N1 (in absolute terms) were positively correlated (high correlation) with the demographic patterns. The % ccH1N1 showed a negative correlation with the population size (Spearman rho: −0.43, *p*: 0.011). A *p*-value higher than the level of marginal significance (i.e., $p > 0.05$) was found analyzing the correlation between the % ccH1N1 and the population density, also the correlation between the % dH1N1 and population density was not significant. It is probable than the federal scale was not suitable for describing the correlation between the variables under investigation. The data at municipal scale is not available.

## The COVID-19 pandemic

Mexican health authorities confirmed the first SARS-CoV-2 positive case in the country at the end of February 2020, a young man who was in CDMX, hospitalized at the National Institute of Respiratory Diseases (INER) that had previously traveled to Italy. Starting from March 2020 the ccCOVID-19 and consequently the victims have increased considerably, placing Mexico among the first places in the unpleasant ranking of countries most affected by COVID-19 pandemic.

The thematic maps (Figs. 3A and 3B) describe the spatial distribution of ccCOVID-19 and dCOVID-19 in Mexico and their association with population density, while the Figs. 4A and 4B show a special focus on CDMX.

The official data of national government (*COVID-19 México, 2020*) indicated 1.2 million ccCOVID-19 and 115.1 thousand dCOVID-19 in Mexico, as of date (18th December, 2020).
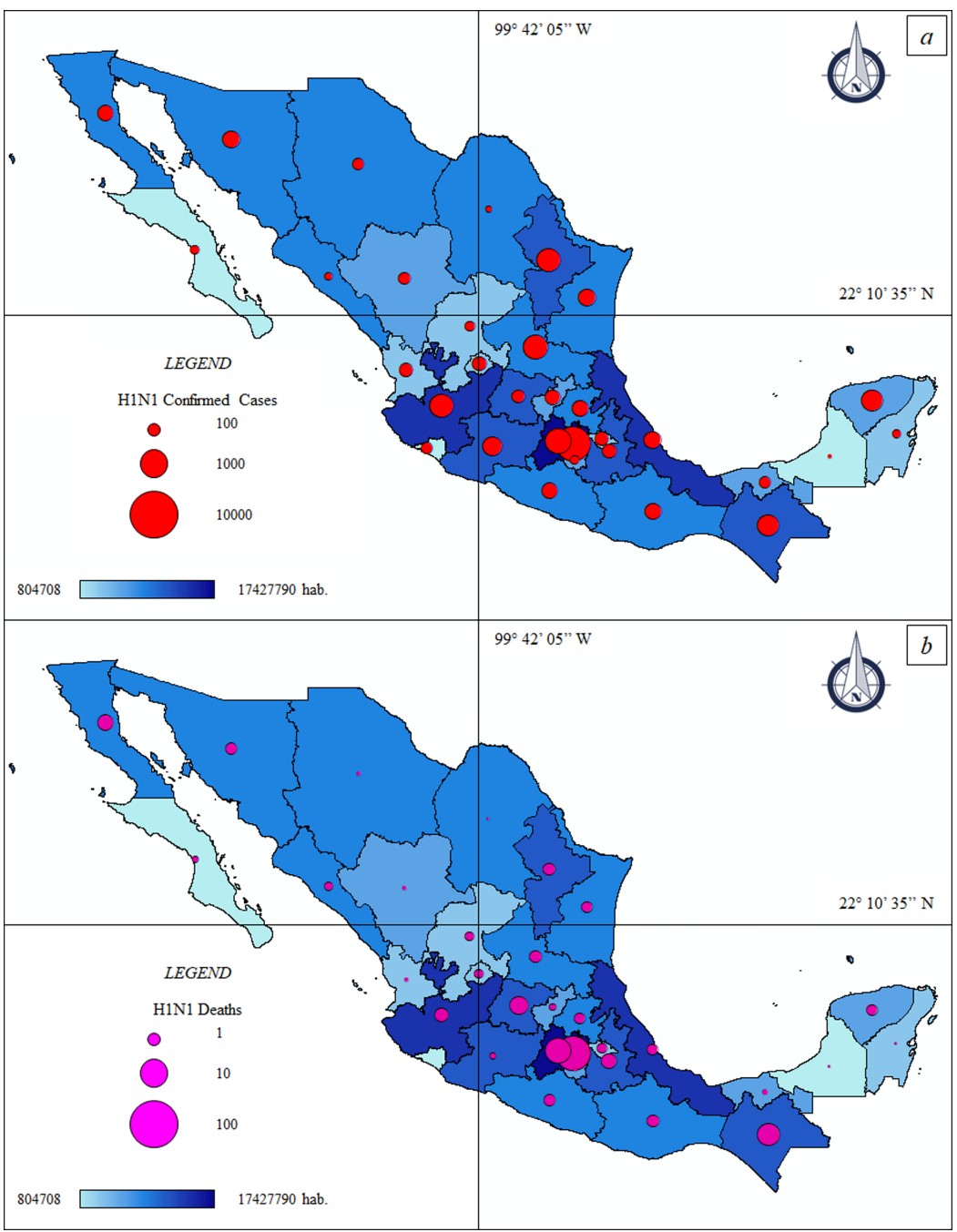

**Figure 1 Thematic maps: Spatial distribution H1N1 pandemic in Mexico—population size.**
(A) Federal H1N1 influenza confirmed cases—population size. ccH1N1: Federal H1N1 influenza confirmed cases (red circle), hab.: Federal population. (B) Federal H1N1 deaths—population size. dH1N1: Federal H1N1 deaths (purple circle), hab.: Federal population.

The CDMX, where 264,330 infections were registered, is the national epicenter of the pandemic (Table 2). The more crowed districts of the capital such as Iztapalapa (37,332 ccCOVID-19–2,812 dCOVID-19), Gustavo A. Madero (31,778 ccCOVID-19–2,570

**Table 1 Distribution of H1N1 pandemic in Mexico—Federal scale.**

| FE | Pop. Size hab. (2010) | Pop. Density hab./km² (2010) | ccH1N1 | dH1N1 | % ccH1N1 | % dH1N1 |
|---|---|---|---|---|---|---|
| Aguascalientes | 1,184,996 | 211 | 1,718 | 48 | 0.14 | 0.0041 |
| Baja California | 3,155,070 | 44 | 2,143 | 47 | 0.07 | 0.0015 |
| Baja California Sur | 637,026 | 9 | 863 | 12 | 0.14 | 0.0019 |
| Campeche | 822,441 | 14 | 219 | 2 | 0.03 | 0.0002 |
| CDMX | 8,851,080 | 5,920 | 8,502 | 152 | 0.10 | 0.0017 |
| Chiapas | 4,796,580 | 65 | 3,711 | 40 | 0.08 | 0.0008 |
| Chihuahua | 3,406,465 | 14 | 1,138 | 44 | 0.03 | 0.0013 |
| Coahuila | 2,748,391 | 18 | 447 | 18 | 0.02 | 0.0007 |
| Colima | 650,555 | 116 | 1,207 | 1 | 0.19 | 0.0002 |
| Durango | 1,632,934 | 13 | 1,298 | 18 | 0.08 | 0.0011 |
| Guanajuato | 5,486,372 | 179 | 1,501 | 52 | 0.03 | 0.0009 |
| Guerrero | 3,388,768 | 53 | 2,057 | 13 | 0.06 | 0.0004 |
| Hidalgo | 2,665,018 | 128 | 2,283 | 44 | 0.09 | 0.0017 |
| Jalisco | 7,350,682 | 94 | 4,333 | 75 | 0.06 | 0.0010 |
| Michoacán | 4,351,037 | 74 | 2,936 | 37 | 0.07 | 0.0009 |
| Morelos | 1,777,227 | 364 | 785 | 22 | 0.04 | 0.0012 |
| Nayarit | 1,084,979 | 39 | 1,675 | 11 | 0.15 | 0.0010 |
| Nuevo León | 4,653,458 | 73 | 4,358 | 76 | 0.09 | 0.0016 |
| Oaxaca | 3,801,962 | 41 | 2,284 | 60 | 0.06 | 0.0016 |
| Puebla | 5,779,829 | 168 | 1,773 | 54 | 0.03 | 0.0009 |
| Querétaro | 1,827,937 | 156 | 1,936 | 37 | 0.11 | 0.0020 |
| Quintana Roo | 1,325,578 | 30 | 728 | 3 | 0.05 | 0.0002 |
| San Luis Potosí | 2,585,518 | 42 | 4,446 | 63 | 0.17 | 0.0024 |
| Sinaloa | 2,767,761 | 48 | 643 | 15 | 0.02 | 0.0005 |
| Sonora | 2,662,480 | 15 | 2,401 | 23 | 0.09 | 0.0009 |
| State of Mexico | 15,175,862 | 679 | 4,682 | 140 | 0.03 | 0.0009 |
| Tabasco | 2,238,603 | 91 | 1,212 | 4 | 0.05 | 0.0002 |
| Tamaulipas | 3,268,554 | 41 | 2,395 | 22 | 0.07 | 0.0007 |
| Tlaxcala | 1,169,936 | 291 | 1,593 | 15 | 0.14 | 0.0013 |
| Veracruz de Ignacio de la Llave | 7,643,194 | 106 | 2,454 | 27 | 0.03 | 0.0004 |
| Yucatan | 1,955,577 | 49 | 3,636 | 24 | 0.19 | 0.0012 |
| Zacatecas | 1,490,668 | 20 | 1,009 | 45 | 0.07 | 0.0030 |
| 95% C.I. | | | | | 0.02 | 0.0003 |
| S.D. | | | | | 0.05 | 0.0008 |

Note:
Pop. Size, Population size; hab., inhabitants; Pop. Density, Population density; ccH1N1, Federal H1N1 confirmed cases; dH1N1, Federal H1N1 deaths; C.I., Confidence Interval; S.D., Standard deviation.

dCOVID-19) and Tlalpan (24,598 ccCOVID-19–765 dCOVID-19) are the principal "red zones" of the country where the pressure on the health system is greatest.

Other FEs that have exceeded the threshold of 40,000 ccCOVID-19 are mainly located in the north—central area of the country or in the Gulf of Mexico, that is, State of Mexico

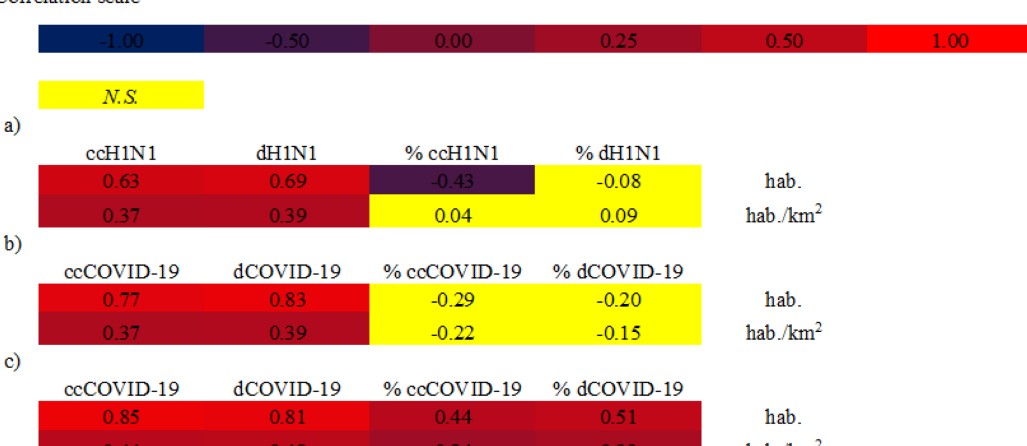

**Figure 2 Spearman correlations between analyzed variables.** (A) Spearman correlation at federal scale —H1N1 pandemic 2009. hab.: Federal population, ccH1N1: Federal H1N1 influenza confirmed cases, dH1N1: Federal H1N1 deaths. (B) Spearman correlation at federal scale—COVID-19 pandemic. hab.: Federal population, ccCOVID-19: Federal COVID-19 confirmed cases, dCOVID-19: Federal COVID-19 deaths. (C) Spearman correlation at municipal scale—COVID-19 pandemic. hab.: Municipal population, ccCOVID-19: Municipal COVID-19 confirmed cases, dCOVID-19: Municipal COVID-19 deaths. Correlations with a $p$ value < 0.01 was considered significant. N.S.: not significant value, $p$ value > 0.01.

(125,628 ccCOVID-19–16,800 dCOVID-19), Nuevo León (73,900 ccCOVID-19–4,810 dCOVID-19), Guanajuato (72,849 ccCOVID-19–4,563 dCOVID-19), Sonora (47,195 ccCOVID-19–3,700 dCOVID-19), Jalisco (46,165 ccCOVID-19–5,287 dCOVID-19), Puebla (42,728 ccCOVID-19–5,350 dCOVID-19), Veracruz de Ignacio de la Llave (41,316 ccCOVID-19–6,105 dCOVID-19) and Tabasco (40,355 ccCOVID-19–3,106 dCOVID-19).

The number of infections and deaths are also alarming in the State of Mexico; the cities of Ecatepec de Morelos (15,684 ccCOVID-19–2,247 dCOVID-19) and Nezahualcóyotl (12,929 ccCOVID-19–1,695 dCOVID-19) connected through a dense public transport network with CDMX are the most exposed areas to the COVID-19 health emergency. In Tlaxcala (9,984 ccCOVID-19–1,314 dCOVID-19), another State close to the capital, the majority of the ccCOVID-19 and dCOVID-19 were recorded in urban areas of Tlaxcala city (1,857 ccCOVID-19–191 dCOVID-19). Approx. two thirds of the ccCOVID-19 in Guerrero (24,723 ccCOVID-19–2,581 dCOVID-19) were found in the Acapulco de Juárez (10,781 ccCOVID-19–1,218 dCOVID-19) and in the neighboring city of Chilpancingo de los Bravo (4,572 ccCOVID-19–276 dCOVID-19), the municipalities of the interior have, at least so far, clearly lower numbers of ccCOVID-19. Yucatan (25,186 ccCOVID-19–2,097 dCOVID-19) showed similar numbers to Guerrero, approx. half of the infections were registered in Merida (15,221 ccCOVID-19–1,175 dCOVID-19). Baja California Sur (15,,809 ccCOVID-19–709 dCOVID-19), Durango (23,301 ccCOVID-19–1,390 dCOVID-19) and Colima (7,534 ccCOVID-19–773 dCOVID-19) were among the FEs of Mexico with less incidence of the COVID-19 pandemic.

The disease fatality rate (quotient between dCOVID-19 and ccCOVID-19) at federal scale (Fig. 5A) ranged between 4.49% of Baja California Sur to 16.01% of Sinaloa.

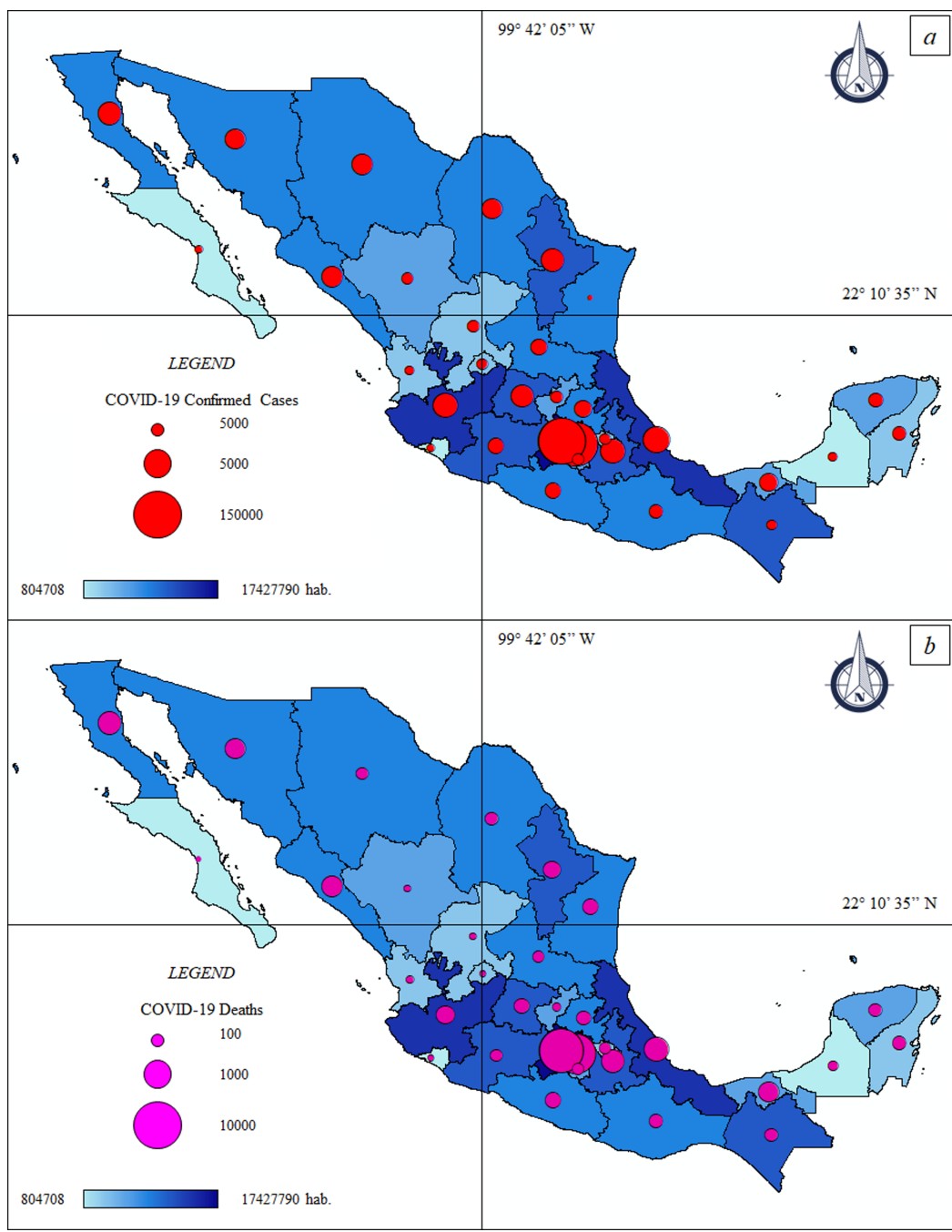

**Figure 3 Thematic maps: Spatial distribution COVID-19 pandemic in Mexico—population size.**
(A) Federal COVID-19 confirmed cases—population size. ccCOVID-19: Federal COVID-19 confirmed cases (red circle), hab.: Federal population. (B) Federal COVID-19 deaths—population size. dCOVID-19: Federal COVID-19 deaths (purple circle), hab.: Federal population.

The Spearman correlation, at federal scale (Fig. 2B), showed different results. On one hand the ccCOVID-19 and dCOVID-19 (in absolute terms) were positively correlated (high correlation) with the population size. On the other hand $p$-values higher than the
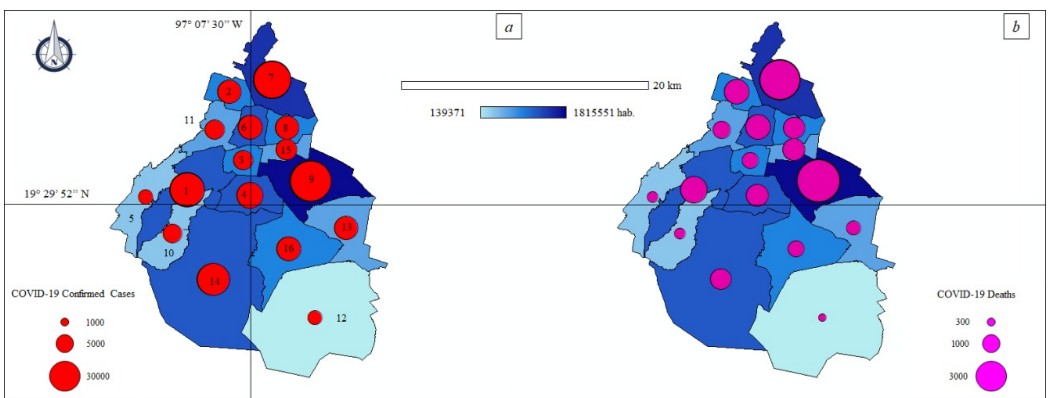

**Figure 4 Thematic maps: Spatial distribution COVID-19 pandemic in Mexico City—population size.** (A) Municipal COVID-19 confirmed cases—population size. ccCOVID-19: Municipal COVID-19 confirmed cases (red circle), hab.: District population. Municipalities of Mexico City. 1: Álvaro Obregón, 2: Azcapotzalco, 3: Benito Juárez, 4: Coyoacán, 5: Cuajimalpa de Morelos, 6: Cuauhtémoc, 7: Gustavo A. Madero, 8: Venustiano Carranza, 9: Iztapalapa, 10: La Magdalena Contreras, 11: Miguel Hidalgo, 12: Milpa Alta, 13: Tláhuac, 14: Tlalpan, 15: Iztacalco, 16: Xochimilco. (B) Municipal COVID-19 deaths—population density. dCOVID-19: Municipal COVID-19 deaths (purple circle), hab.: District population.

level of marginal significance (i.e., $p > 0.05$) were found analyzing the correlations between the % ccCOVID-19 and the demographic patterns; also the correlations between the % dH1N1 and the demographic patterns were not significant. These results lead us to suppose that the federal scale do not allow to observe the correlation between the pandemics indicators and the two demographic patterns, therefore the next step was to choose a more detailed geographical scale, that is, on a municipal basis.

The Spearman analysis, at municipal scale (Fig. 2C) (2,455 municipalities), showed high correlation between the data. The ccCOVID-19 and dCOVID-19 (in absolute terms) were high positively correlated with the demographic patterns; also the % ccCOVID-19 and % dCOVID-19 showed moderate high correlations with demographic patterns.

The municipal scale resulted more suitable than the federal scale to describe the spread of COVID-19 pandemic. The $R^2$ of LOESS (Fig. 6A) was equal to 0.76 indicating medium high correlation between ccCOVID-19 and municipal population, while the $R^2$ of 0.91 indicated high correlation between dCOVID-19 and the municipal population (Fig. 6B). However, it is important to emphasize that correlation is not causation and it is not right to say that high population sizes caused more dCOVID-19 despite the analysis regression showed that the variables were related to each other's.

The urban areas, that is, federal capitals and principal municipalities, were particularly affected by the COVID-19 pandemic; some example: in the city of La Paz, where approximately the 37.52% of the total Baja California Sur' population lives, the 50.22% of federal ccCOVID-19 were registered; Ciudad Juárez, the biggest city of Chihuahua with the 44.06% of federal population, showed the 53.81% of federal ccCOVID-19; also in the port city of Veracruz, one of the most important trading centers in the Gulf of Mexico where the 7.34% of federal population lives, the federal ccCOVID-19 were the 21.56%.

**Table 2 Distribution of COVID-19 pandemic in Mexico—Federal scale.**

| FE | Pop. Size hab. (2020) | Density Pop. hab./km² (2020) | ccCOVID-19 | dCOVID-19 | % ccCOVID-19 | % dCOVID-19 |
|---|---|---|---|---|---|---|
| Aguascalientes | 1,434,635 | 255 | 15,683 | 1,218 | 1.09 | 0.08 |
| Baja California | 3,634,868 | 51 | 29,942 | 4,783 | 0.82 | 0.13 |
| Baja California Sur | 804,708 | 11 | 15,809 | 709 | 1.96 | 0.09 |
| Campeche | 1,000,617 | 17 | 6993 | 946 | 0.70 | 0.09 |
| CDMX | 9,018,645 | 6,033 | 264,330 | 14,733 | 2.93 | 0.16 |
| Chiapas | 5,730,367 | 78 | 7,963 | 1,177 | 0.14 | 0.02 |
| Chihuahua | 3,801,487 | 15 | 33,669 | 4,086 | 0.89 | 0.11 |
| Coahuila | 3,218,720 | 21 | 45,280 | 3,755 | 1.41 | 0.12 |
| Colima | 785,153 | 140 | 7,534 | 773 | 0.96 | 0.10 |
| Durango | 1,868,996 | 15 | 23,301 | 1,390 | 1.25 | 0.07 |
| Guanajuato | 6,228,175 | 203 | 72,849 | 4,563 | 1.17 | 0.07 |
| Guerrero | 3,657,048 | 58 | 24,723 | 2,581 | 0.68 | 0.07 |
| Hidalgo | 3,086,414 | 148 | 21,396 | 2,988 | 0.69 | 0.10 |
| Jalisco | 8,409,693 | 107 | 46,165 | 5,287 | 0.55 | 0.06 |
| Michoacán | 4,825,401 | 82 | 30,378 | 2,489 | 0.63 | 0.05 |
| Morelos | 2,044,058 | 419 | 8,642 | 1,356 | 0.42 | 0.07 |
| Nayarit | 1,288,571 | 46 | 7,518 | 1,000 | 0.58 | 0.08 |
| Nuevo León | 5,610,153 | 87 | 73,900 | 4,810 | 1.32 | 0.09 |
| Oaxaca | 4,143,593 | 44 | 26,011 | 2,009 | 0.63 | 0.05 |
| Puebla | 6,604,451 | 193 | 42,728 | 5,350 | 0.65 | 0.08 |
| Querétaro | 2,279,637 | 195 | 27,036 | 1,641 | 1.19 | 0.07 |
| Quintana Roo | 1,723,259 | 39 | 15,057 | 1,978 | 0.87 | 0.11 |
| San Luis Potosí | 2,866,142 | 47 | 37,412 | 2,787 | 1.31 | 0.10 |
| Sinaloa | 3,156,674 | 54 | 25,447 | 4,093 | 0.81 | 0.13 |
| Sonora | 3,074,745 | 17 | 47,195 | 3,700 | 1.53 | 0.12 |
| State of Mexico | 17,427,790 | 780 | 125,628 | 16,800 | 0.72 | 0.10 |
| Tabasco | 2,572,287 | 104 | 40,355 | 3,106 | 1.57 | 0.12 |
| Tamaulipas | 3,650,602 | 45 | 37,343 | 3,114 | 1.02 | 0.09 |
| Tlaxcala | 1,380,011 | 344 | 9,984 | 1,314 | 0.72 | 0.10 |
| Veracruz de Ignacio de la Llave | 8,539,862 | 119 | 41,316 | 6,105 | 0.48 | 0.07 |
| Yucatan | 2,259,098 | 57 | 25,186 | 2,097 | 1.11 | 0.09 |
| Zacatecas | 1,666,426 | 22 | 19,201 | 1,560 | 1.15 | 0.09 |
| 95% C.I. | | | | | 0.19 | 0.01 |
| S.D. | | | | | 0.52 | 0.03 |

**Note:**
Pop. Size, Population size; hab., inhabitants; Pop. Density, Population density; ccCOVID-19, Federal COVID-19 confirmed cases; dCOVID-19, Federal COVID-19 deaths; C.I., Confidence Interval; S.D., Standard deviation.

The discrepancy between the % of the federal population and the % of federal ccCOVID-19 was visible in most cases throughout all of Mexico (Table 3): Aguascalientes (Aguascalientes), Mexicali (Baja California), San Francisco de Campeche (Campeche), Tuxtla Gutiérrez (Chiapas), Colima (Colima), Victoria de Durango (Durango), León

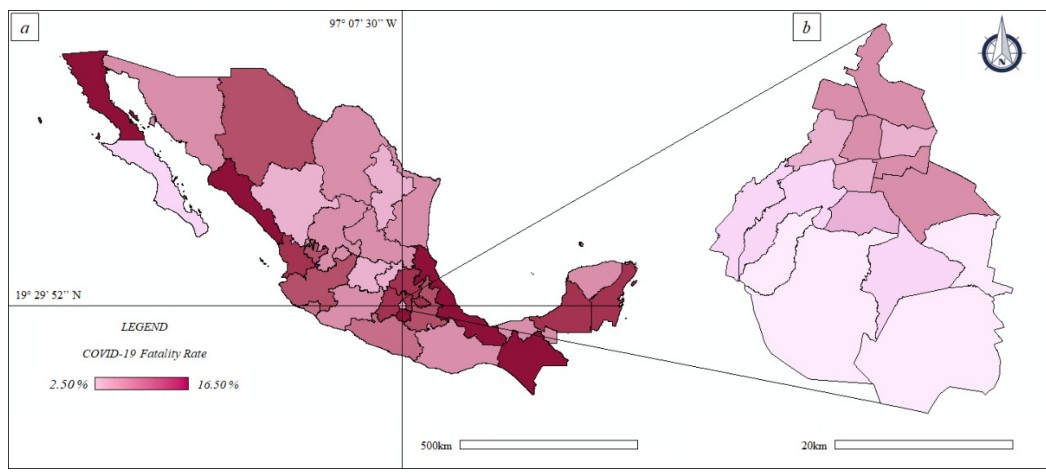

**Figure 5 Thematic maps: spatial distribution COVID-19 fatality rate in Mexico and Mexico City—population size.** (A) Mexico. (B) Mexico City.

(Guanajuato), Acapulco de Juárez and Chilpancingo de los Bravo (Guerrero), Pachuca de Soto (Hidalgo), Guadalajara (Jalisco), Morelia (Michoacán), Cuernavaca (Morelos), Tepic (Nayarit), Monterrey (Nuevo León), Oaxaca de Juárez (Oaxaca), Puebla de Zaragoza (Puebla), Santiago de Querétaro (Querétaro), Cancun—Benito Juárez and Chetumal—Othón P. Blanco (Quintana Roo), San Luis Potosí (San Luis Potosí), Culiacán Rosales (Sinaloa), Hermosillo (Sonora), Ecatepec de Morelos, Nezahualcóyotl and Toluca de Lerdo (State of Mexico), Villahermosa—Centro (Tabasco), Ciudad Victoria (Tamaulipas), Tlaxcala (Tlaxcala), Merida (Yucatan), Zacatecas (Zacatecas). The discrepancy between the previously indicated values should not exist under uniform conditions, probably it could be traced back by the incidence of key factors such as population size and density.

The disease fatality rate, at municipal basis ranged until approx. 20.00% (Baja California); values of approximately 14.00% were found in many cities of Mexico including Cuernavaca (Morelos), Culiacan (Sinaloa), Ecatepec de Morelos and Nezahualcoyotl (State of Mexico), Veracruz (Veracruz de Ignacio de la Llave). In CDMX the disease fatality rate (Fig. 5B) ranged between 2.50% of Milpa Alta and approx. 7.50% of the Iztapalapa and Iztacalco.

# DISCUSSION

## The pandemics in Mexico

The COVID-19 pandemic is increasing worldwide, with the American continent as one of its epicenters and Mexico is among the ten worst countries in terms of infections and deaths.

The magnitude of COVID-19 pandemic is much higher than the "Swine flu" pandemic; nevertheless it is possible to identify common aspects and differences between the two health emergencies. As in 2009 the most populated FEs of Mexico are the "red spots of infection" and report the greatest number of confirmed cases and deaths.

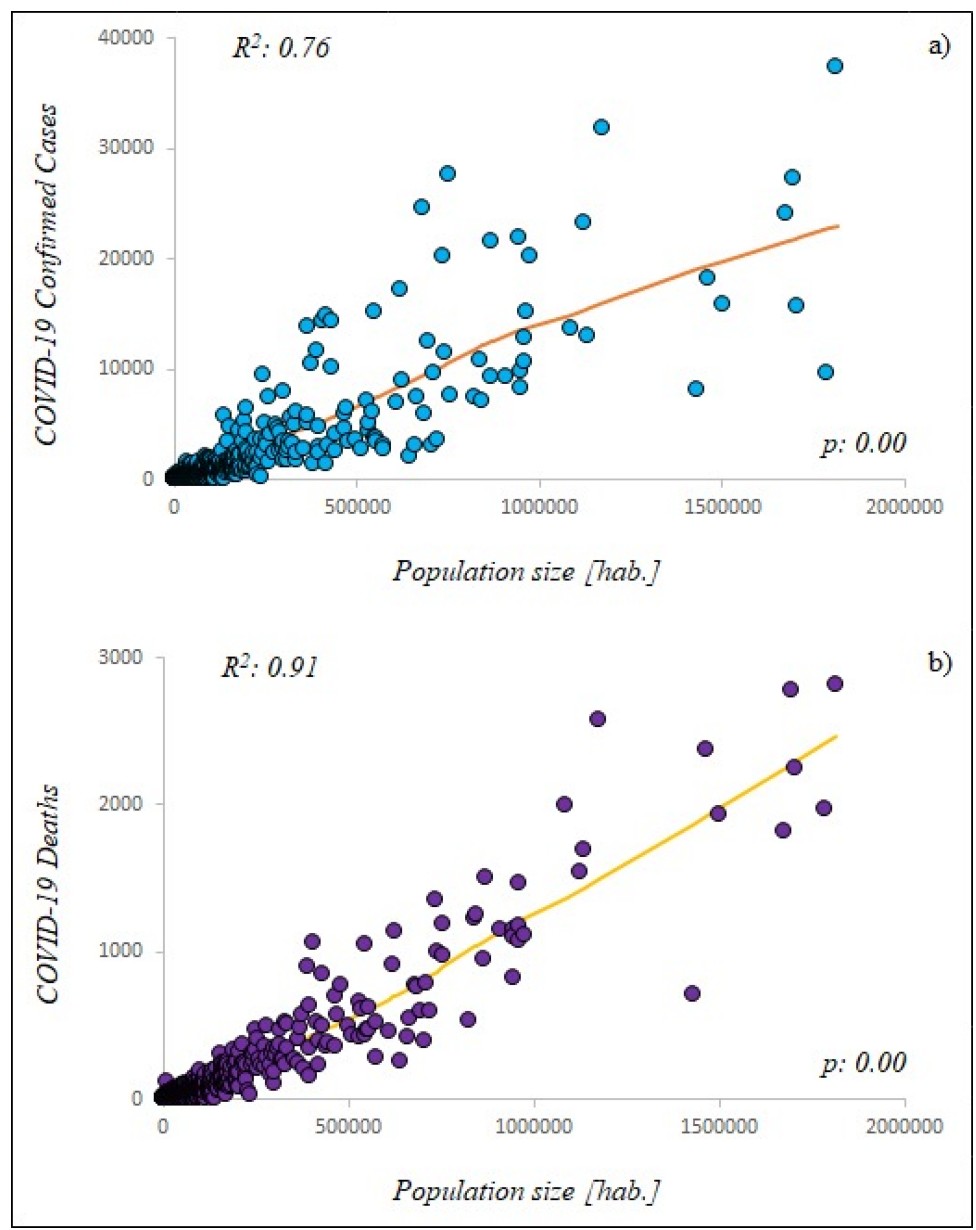

**Figure 6 Locally weighted linear regression (LOESS) between studied variables.** (A) Municipal scale: COVID-19 confirmed cases—population size (blue circle), (B) Municipal scale: COVID-19 deaths—population size (purple circle). hab.: Municipal population.

The authorities have been trying to reduce the spread of the COVID-19 by mitigation activities and strategies such as social distancing, travel restrictions, closing schools, shutting down nonessential activities and quarantine (*Méndez-Arriaga, 2020*), as they had previously done during the 2009 health emergency (*Chowell et al., 2008*). The major trouble has been identifying the root and the new cluster of infection in the urban areas.

The SARS-CoV-2 is transmitted by human-to-human, symptomatic people are the most frequent source of contagious; the virus can also be spread by contaminated objects

**Table 3 Distribution of COVID-19 in the principal municipalities or districts of Mexico—Municipal scale.**

| Federal Entities | Major municipalities or districts | Population | % Population (Federal Entities) | Confirmed COVID-19 Cases | % Confirmed COVID-19 Cases (Federal Entities) | Confirmed COVID-19 Deaths | % Confirmed COVID-19 Deaths (Federal Entities) |
|---|---|---|---|---|---|---|---|
| Aguascalientes | | 1,434,635 | | 15,683 | | 1,218 | |
| | Aguascalientes | 961,977 | 67.05 | 12,730 | 81.17 | 1,077 | 88.42 |
| Baja California | | 3,634,868 | | 29,942 | | 4,783 | |
| | Mexicali | 1,087,478 | 29.92 | 13,624 | 45.50 | 1,996 | 41.73 |
| | Tijuana | 1,789,531 | 49.23 | 9,585 | 32.35 | 1,973 | 41.25 |
| Baja California Sur | | 804,708 | | 15,809 | | 709 | |
| | La Paz | 301,961 | 37.52 | 7,939 | 50.22 | 343 | 48.38 |
| Campeche | | 1,000,617 | | 6,993 | | 946 | |
| | San Francisco de Campeche | 317,424 | 31.72 | 2,909 | 41.60 | 370 | 39.11 |
| Chiapas | | 5,730,367 | | 7,963 | | 1,177 | |
| | Tuxtla Gutiérrez | 662,591 | 11.56 | 2,970 | 37.30 | 420 | 35.68 |
| Chihuahua | | 3,801,487 | | 33,669 | | 4,086 | |
| | Chihuahua | 949,395 | 24.97 | 8,194 | 24.34 | 824 | 20.17 |
| | Ciudad Juárez | 1,674,973 | 44.06 | 18,116 | 53.81 | 2,366 | 57.91 |
| Coahuila de Zaragoza | | 3,218,720 | | 45,280 | | 3,755 | |
| | Saltillo | 869,184 | 27.00 | 9,269 | 20.47 | 950 | 25.30 |
| Colima | | 785,153 | | 7,534 | | 773 | |
| | Colima | 169,188 | 21.55 | 2,013 | 26.72 | 177 | 22.90 |
| Durango | | 1,868,996 | | 23,301 | | 1,390 | |
| | Victoria de Durango | 654,876 | 35.04 | 12,393 | 53.19 | 588 | 42.30 |
| Guanajuato | | 6,228,175 | | 72,849 | | 4,563 | |
| | Guanajuato | 198,035 | 3.18 | 2,620 | 3.60 | 132 | 2.89 |
| | León | 1,679,610 | 26.97 | 24,126 | 33.12 | 1,814 | 39.75 |
| Guerrero | | 3,657,048 | | 24,723 | | 2,581 | |
| | Acapulco de Juárez | 840,795 | 22.99 | 10,781 | 43.61 | 1,218 | 47.19 |
| | Chilpancingo de los Bravo | 284,330 | 7.77 | 4,572 | 18.49 | 276 | 10.69 |
| Hidalgo | | 3,086,414 | | 21,396 | | 2,988 | |
| | Pachuca de Soto | 280,312 | 9.08 | 4,817 | 22.51 | 499 | 16.70 |
| Jalisco | | 8,409,693 | | 46,165 | | 5,289 | |
| | Guadalajara | 1,503,505 | 17.88 | 15,837 | 34.31 | 1,930 | 36.49 |
| CDMX | | 9,018,645 | | 264,330 | | 14,733 | |
| | Álvaro Obregón | 755,537 | 8.38 | 27,558 | 10.43 | 1,191 | 8.08 |
| | Azcapotzalco | 408,441 | 4.53 | 14,294 | 5.41 | 1,058 | 7.18 |
| | Benito Juárez | 433,708 | 4.81 | 10,129 | 3.83 | 499 | 3.39 |
| | Coyoacán | 621,952 | 6.90 | 17,076 | 6.46 | 906 | 6.15 |
| | Cuajimalpa de Morelos | 199,809 | 2.22 | 6,324 | 2.39 | 251 | 1.70 |
| | Cuauhtémoc | 548,606 | 6.08 | 15,131 | 5.95 | 1,050 | 7.13 |
| | Gustavo A. Madero | 1,176,967 | 13.05 | 31,778 | 12.02 | 2,570 | 17.44 |

*(Continued)*

| Federal Entities | Major municipalities or districts | Population | % Population (Federal Entities) | Confirmed COVID-19 Cases | % Confirmed COVID-19 Cases (Federal Entities) | Confirmed COVID-19 Deaths | % Confirmed COVID-19 Deaths (Federal Entities) |
|---|---|---|---|---|---|---|---|
| | Iztacalco | 393,821 | 4.37 | 11,598 | 4.39 | 895 | 6.07 |
| | Iztapalapa | 1,815,551 | 20.13 | 37,332 | 14.12 | 2,812 | 19.09 |
| | La Magdalena Contreras | 245,147 | 2.72 | 9,429 | 3.57 | 254 | 1.72 |
| | Miguel Hidalgo | 379,624 | 4.21 | 10,511 | 3.98 | 574 | 3.90 |
| | Milpa Alta | 139,371 | 1.55 | 5,803 | 2.23 | 148 | 1.00 |
| | Tláhuac | 366,586 | 4.06 | 13,798 | 5.22 | 402 | 2.73 |
| | Tlalpan | 682,234 | 7.56 | 24,598 | 9.31 | 765 | 5.19 |
| | Venustiano Carranza | 433,231 | 4.80 | 14,231 | 5.38 | 843 | 5.72 |
| | Xochimilco | 418,060 | 4.64 | 14,734 | 5.57 | 515 | 3.50 |
| Michoacán | | 4,825,401 | | 30,378 | | 2,489 | |
| | Morelia | 825,585 | 17.11 | 7,391 | 24.33 | 530 | 21.29 |
| Morelos | | 2,044,058 | | 8,642 | | 1,356 | |
| | Cuernavaca | 399,426 | 19.54 | 2,424 | 28.05 | 342 | 25.22 |
| Nayarit | | 1,288,571 | | 7,518 | | 1,000 | |
| | Tepic | 445,889 | 34.60 | 3,996 | 53.15 | 375 | 37.50 |
| Nuevo León | | 5,610,153 | | 73,900 | | 4,810 | |
| | Monterrey | 1,124,835 | 20.05 | 23,194 | 31.39 | 1,541 | 32.04 |
| Oaxaca | | 4,143,593 | | 26,011 | | 2,009 | |
| | Oaxaca de Juárez | 258,636 | 6.24 | 7,479 | 28.75 | 405 | 20.16 |
| Puebla | | 6,604,451 | | 42,728 | | 5,350 | |
| | Puebla de Zaragoza | 1,698,509 | 25.72 | 27,301 | 63.89 | 2,777 | 51.91 |
| Querétaro | | 2,279,637 | | 27,036 | | 1,641 | |
| | Santiago de Querétaro | 626,495 | 27.48 | 20,233 | 74.84 | 1,117 | 68.07 |
| Quintana Roo | | 1,723,259 | | 15,057 | | 1,978 | |
| | Cancun (Benito Juárez) | 743,626 | 43.15 | 6,989 | 46.42 | 1,250 | 63.20 |
| | Chetumal (Othón P. Blanco) | 265,298 | 15.40 | 3,975 | 26.40 | 226 | 11.43 |
| San Luis Potosí | | 2,866,142 | | 37,412 | | 2,787 | |
| | San Luis Potosí | 870,578 | 30.37 | 21,482 | 57.42 | 1,501 | 53.86 |
| Sinaloa | | 3,156,674 | | 25,447 | | 4,093 | |
| | Culiacán Rosales | 962,871 | 30.50 | 10,668 | 41.92 | 1,467 | 35.84 |
| Sonora | | 3,074,745 | | 47,195 | | 3,700 | |
| | Hermosillo | 946,054 | 30.77 | 21,830 | 46.25 | 1,144 | 30.92 |
| State of Mexico | | 17,427,790 | | 125,628 | | 16,800 | |
| | Ecatepec de Morelos | 1,707,754 | 9.80 | 15,684 | 12.48 | 2,247 | 13.38 |
| | Nezahualcóyotl | 1,135,786 | 6.52 | 12,929 | 10.29 | 1,695 | 10.09 |
| | Toluca de Lerdo | 948,950 | 5.45 | 9,716 | 7.73 | 1,101 | 6.55 |
| Tabasco | | 2,572,287 | | 40,355 | | 3,106 | |
| | Villahermosa (Centro) | 739,611 | 28.75 | 20,132 | 49.89 | 1,356 | 43.66 |

| Table 3 (continued) | | | | | | | |
|---|---|---|---|---|---|---|---|
| Federal Entities | Major municipalities or districts | Population | % Population (Federal Entities) | Confirmed COVID-19 Cases | % Confirmed COVID-19 Cases (Federal Entities) | Confirmed COVID-19 Deaths | % Confirmed COVID-19 Deaths (Federal Entities) |
| Tamaulipas | | 3,650,602 | | 37,343 | | 3,114 | |
| | Ciudad Victoria | 367,051 | 10.05 | 5,116 | 13.70 | 237 | 7.61 |
| | Reynosa | 686,670 | 18.81 | 5,931 | 15.88 | 754 | 24.21 |
| Tlaxcala | | 1,380,011 | | 9,984 | | 1,314 | |
| | Tlaxcala | 103,435 | 7.50 | 1,857 | 18.60 | 191 | 14.54 |
| Veracruz de Ignacio de la Llave | | 8,539,862 | | 41,316 | | 6,105 | |
| | Veracruz | 626,918 | 7.34 | 8,906 | 21.56 | 1,142 | 18.71 |
| | Xalapa - Enríquez | 513,443 | 6.01 | 2,662 | 6.44 | 434 | 7.11 |
| Yucatán | | 2,259,098 | | 25,186 | | 2,097 | |
| | Merida | 963,861 | 42.67 | 15,221 | 60.43 | 1,175 | 56.03 |
| Zacatecas | | 1,666,426 | | 19,201 | | 1,560 | |
| | Zacatecas | 155,533 | 9.33 | 4,762 | 20.80 | 305 | 19.55 |
| 95% C.I. | | | 3.88 | | 5.51 | | 5.44 |
| S.D. | | | 14.49 | | 20.58 | | 20.33 |

Note:
C.I., Confidence Interval; S.D., Standard deviation.

and aerosol inhalation (*Helmy et al., 2020*). The metropolitan areas, involving districts tightly linked through socio-economic relationships and permanent commuting of citizens, workers and tourists are more exposed to the pandemic outbreaks compared to rural areas (*Hamidi, Sabouri & Ewing, 2020*). Several causes could explain the correlation, observed at municipal scale, between demographic factors and the COVID-19 spread. The urban areas or those with high population size and density imply that more people live in per unit of area increasing the direct and indirect contact among the citizens, workers and tourist (*Macharia, Joseph & Okiro, 2020*). *Chhikara et al. (2020)* indicate that high population density increased the possibility of COVID-19 diffusion, the urban-rural segmentation reduced the spread of outbreaks between urban and rural areas.

*Méndez-Arriaga (2020)*, investigating environmental climate patterns of COVID-19 spread in Mexico, showed that the highest national local transmission in CDMX and State of Mexico were not only the result of the environmental factors (temperature and regional climate) but also the demographic characteristics (population size and density).

According to *Chowell et al. (2011)*, the three H1N1 pandemic waves across Mexican FEs were driven by the most populated cities, the earlier onset occurred in the overcrowded FEs located in the central part of the country, as it is nowadays during the COVID-19 emergency; however, in their publication the authors indicated that incidence rates observed in large population centers was lowest if compared with rural zone. The rural zones, as well as the degraded peripheral areas, in most cases, have structural gaps that is,

small and poorly efficient hospitals, precarious sanitary conditions that could accelerate the spread of infectious diseases.

In regard to COVID-19 pandemic, several countries are facing a second wave of infections, however, Mexico has not been the case yet and this study does not cover that scenario.

## Sanitary emergencies in CDMX

The CDMX is the national epicenter point of the actual outbreak; all its citizens, that live closely in a "reduced space", appear to have accelerated the virus spread through multiple contacts and interactions in packed subway trains, busy playgrounds and crowded residential buildings creating "red spots" of infections.

In particular, the districts of Iztapalapa and Gustavo A. Madero had the highest incidence of COVID-19 confirmed cases at national scale.

According to *Zepeda-Lopez et al. (2010)*, also during the outbreak of the novel influenza A (H1N1)v virus in 2009, CDMX showed the higher number of ccH1N1 principally in the crowed districts of Iztapalapa and Gustavo A. Madero. According to *Ponnambalam et al. (2012)* and *You, Wu & Guo (2020)*, population density, size if population and urban land are key factors in the infectious diseases transmission and, consequently, the urban areas present a significant challenge for national public health.

The underground (metro), with approximately 4.5 million commuters every day (1.6 billion ridership per year), is the most crowded urban transportation system of CDMX. It brings people in close contact and facilitates the virus spread by human-to-human transmission. During the daily massive movements, the passengers, sharing the same elements (surfaces, objects) and air, promoted a significant interchange of human or environmental microbiota causing the wider spread of pathogens (*Vargas-Robles et al., 2020*). The terminal stations such as Indios Verdes in the delegation of Gustavo A. Madero (north of CDMX) and Pantitlan in Venustiano Carranza (east of CDMX) that connect the inner city to suburban areas of Ecatepec de Morelos and Nezahualcóyotl (State of Mexico) were considered among the main "red spots" of COVID-19 contagious in CDMX by national government.

In contrast to the policies implemented by most of countries worldwide, Mexico in March 2020 preferred not to apply restrictions for air travel keeping the Benito Juárez International Airport of CDMX opened, that is the nation's main gateway with approximately 50 million commuters every year. Foreign tourists or Mexican citizens returning may have been vectors of the virus.

Massive religious events, such as the representation of the crucifixion of Jesus in Iztapalapa (Iztapalapa Passion Play 2009), which was held few days before the identification of the virus A (H1N1) pdm09 and was attended by more than 2 million people, were indeed dangerous events for the spread of the virus at that time. According to *Zepeda-Lopez et al. (2010)*, the congregation of people in the Iztapalapa Passion Play has contributed to the roll out of the virus A (H1N1) pdm09 during the initial outbreak phase throughout CDMX and beyond. Even more, while music events such as Estéreo Picnic, Lollapalooza Chile, Argentina y Brazil were reprogramed for next year, the

"Vive Latino", one of the most important Rock festivals in Latin America took place in the midst of COVID-19 pandemic, on 14th and 15th March at the Foro Sol Stadium in CDMX, gathering more of 40,000 people. The government and festival organization, instead of suspending the musical show and postpone it on another occasion, have chosen to take health preventive measures installing eight medical tents, filters with infrared thermometers at the entrances, and a team of 92 paramedics, 10 doctors and 8 ambulances (*Salud, 2020*).

The government of CDMX identified seven market places including "Central de Abasto" in Iztapalapa and "La Merced" in Cuauhtémoc as significant sources of infection. They recommended the citizens to avoid buying groceries in those places. In particular, the "Central de Abasto", that supplies the capital and the center of the country and receives perishable food from all of Mexico, is an agglomeration point in the city representing high risk infection. The entry at this place was prohibited to pregnant women, children and older adults and the use of facemask mandatory.

Misleading statements by government representatives and the lack of official restrictions that is, other than the closure of schools and suspension of classes in all academic levels, had left many of the preventative measures to citizen's "good judgment", since self isolation was voluntary, in the poorest areas of the CDMX such as Iztapalapa where several thousands of informal workers that have no other means of subsistence, and live on a "day to day" basis, need to work daily to meet basic family needs putting their health and their families at risk. Hence the decision to close the capital's "Zócalo" starting from the last week of March with almost 270 metal fences that protected the place in order to avoid crowds, potential meetings or social outings.

Public and private universities in CDMX have taken mitigation measures against the strain of COVID-19, suspending face-to-face activities and starting online programs. *Chowell et al. (2011)* analyzing the epidemiological patterns of "Swine flu" pandemic highlighted the importance of school closure and other restriction measures to mitigate the impact of "Swine flu" pandemic and future sanitary emergencies in urban and rural areas.

High densely populated cities within the same country have shown a grater out brake of COVID-19 than less densely populated cities (*Tira, 2020*). However, it is important to mention the limitations of the observations presented here since the analysis were made using phew parameters that is, population, km$^2$, population density, ccCOVID-19 and dCOVID-19; several other studies have shown the need to focus also on weather differences among regions (*Rashed et al., 2020*), seasonality (*Smit et al., 2020*; *Carlson et al., 2020*) and social parameters such as income (*Bonacini, Gallo & Scicchitano, 2020*).

## CONCLUSIONS

The geographical scale results a key factor to accurately describe the spread of viral diseases in Mexico; an adequate geographical scale of reference is crucial for designing more efficient control or mitigation measures in such a heterogeneous and socially complex population during pandemic emergencies. The municipality scale that used of 2,455 data against the 32 data of federal scale decreasing the uncertainty of the analysis and confirmed

the correlation between demographic parameters and pandemic indicators; in our research a detailed geographic scale has produced better correlates with available data and highlighted the areas most exposed to the health emergency such us the delegation of Iztapalapa and Gustavo A. Madero in CDMX.

## ACKNOWLEDGEMENTS

The authors thank one anonymous reviewer for helpful comments on a previous version of the manuscript.

### Funding

The authors received no funding for this work.

### Competing Interests

The authors declare that they have no competing interests.

### Author Contributions

- Yohanna Sarria-Guzmán conceived and designed the experiments, performed the experiments, prepared figures and/or tables, authored or reviewed drafts of the paper, searched the official data in the specific sources, and approved the final draft.
- Jaime Bernal performed the experiments, authored or reviewed drafts of the paper, and approved the final draft.
- Michele De Biase performed the experiments, analyzed the data, authored or reviewed drafts of the paper, and approved the final draft.
- Ligia C. Muñoz-Arenas analyzed the data, authored or reviewed drafts of the paper, and approved the final draft.
- Francisco Erik González-Jiménez performed the experiments, prepared figures and/or tables, authored or reviewed drafts of the paper, and approved the final draft.
- Clemente Mosso analyzed the data, authored or reviewed drafts of the paper, and approved the final draft.
- Arit De León-Lorenzana analyzed the data, prepared figures and/or tables, authored or reviewed drafts of the paper, and approved the final draft.
- Carmine Fusaro conceived and designed the experiments, performed the experiments, analyzed the data, prepared figures and/or tables, authored or reviewed drafts of the paper, and approved the final draft.

### Data Availability

Raw data are available as a Supplemental File.

### Supplemental Information

Supplemental information for this article can be found online at http://dx.doi.org/10.7717/peerj.11144#supplemental-information.

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
