# Peer review of "Using demographic data to understand the distribution of H1N1 and COVID-19 pandemics cases among federal entities and municipalities of Mexico"

_PeerJ, doi:10.7717/peerj.11144_

## Round 0.1 · original submission · Major Revisions

I am in agreement with the experts who reviewed and provided their comments and edits on your manuscript. My main concern, like them, is that there could have been much more spatial analysis beyond simple correlations (specially that the correlations between cases and dealths of the same pandemic is not really the primarily interest here). Because your choice of correlating both pandemics is very noble, I would like to give you a chance to respond to the reviewers, add analysis comparing both pandemics (this will exponentially increase the interest in your research).

Reviewer 1 ·

Basic reporting

1. I am not sure of the referencing style used, where in most cases a URL has been provided and a year given in the manuscript. It should be the author/organization and year

2. Review of the risk factors of the pandemic not sufficient

3. The title does not reflect the analysis conducted

4. The current paper would fit better into a narrative style for COVID-19, otherwise more detailed analysis needed

Experimental design

1. It's not clear what the first objective is when the author says analyse. This should be phrased more clearly.

2. Line 125, what do cc and d mean ( prefixes in COVID-19) ccCOVID-19, dCOVID-19. Did the authors mean comparisons of COVID-19 cases and deaths, then write in full before abbreviating. The same applies to ccH1N1 and dH1N1


3. Line 136, how were the factors arrived at? Many factors would have been considered, why the specific ones listed and not others. In literature, there have been many risk factors of COVID-19 including for severe cases and deaths.

4. What does (hab.) mean? If it’s the unit of measurement, write in full then in brackets the abbreviation. The same applies to the other units of measurements used in the manuscript

5. The methods are not clear. Please tabulate all the factors considered after clearly indicating the rationale for their inclusions. Why was correlation chosen as the measure of association? Why did the authors not use regression analysis where they would also control for cofounders and spatial autocorrelation?

6. The title Mapping the clustering of people to further understand the spread of viral infections is not appropriate given the evidence presented. There was no clustering analysis presented or any evidence on the spread of viral infections.

7. The current analysis has looked only into correlations between the pandemics and randomly chosen factors among many risk factors. The title should be paraphrased to reflect the analysis done. Or add more analysis such as clustering to substantiate the title.

8. Line 157-181, these are not results but country context details. Add a section at the beginning of the methods called country context. Some maps would help visualize these data in the country context section.

9. Line 183 to 202, the details presented are mainly describing the pandemic with few results or correlation. Would the authors consider this being a description of the two pandemics as opposed to analysis paper? The details presented would fit better into a paper describing how the COVID-19 ( exclude H1N1) unravelled in Mexico and the control measures the government has been putting into place.

10. Further, why investigate the correlation between deaths and cases of the same pandemic? A death will be a case in the first instance. Further, there are different riks factors of deaths and different risk factors of the cases, therefore the variables should have been separated to those that are associated with a death and those associated with a case. These can be mined from the literature of the two pandemics. The same applies to the COVID-19 description

11. The authors should note that correlation does not refer to causation. For example, in line 247, the authors are implying causation which is not quite right

12. The cases presented in the maps should also be corrected for the population. The more populate areas will have cases, therefore dividing by the population gives a better measure per geographical unit

Validity of the findings

The discussion, methods and results can be combined into a narrative/descriptive paper describing how the COVID-19 ( exclude H1N1) unravelled in Mexico and the control measures the government has been putting into place.

·

Basic reporting

The article is written in clear, unambiguous, and technically correct language.
The introduction and background revealed the impact of the disease and the relevance of the analysis performed. some edits are suggested.

I find confusing when the authors describe percentages using “% X/hab.” (As done for the variables ‘%ccH1N1/hab.”, “dH1N1/hab.”, “%ccCOVID-19/hab.” and “%dCOVID-19/hab.”) I would remove the "/hab." from the names it since it could be misinterpreted and you are already describing that you are using a percentage of the total population (i.e. using the name “%ccCOVID-19” instead of "%ccCOVID-19/hab.").

Raw data has duplicated columns per sheet, non descriptive column names, and not well labeled for the disease outcomes analyzed (it doesn't specify which table is for which disease). Providing a data dictionary and cleaning it a bit more is suggested.

Experimental design

The aims were well established, and the presented results describe well the relationship between population density, and case counts and deaths.

I don’t see the point of getting a correlation between the case counts and deaths since this just describes the same as disease specific mortality, which I think it would be more interesting to look at and can provide more succinct and complete information. I suggest the authors to use disease specific mortality to describe their results on the lines 213-239 and include it in the tables and thematic maps.

Validity of the findings

Underlying data have been provided and conclusions were limited to supporting results.
Something that is not mentioned and I think is very relevant to the conclusion on the geographical scale, is that when you've from the federal level to municipal, you are effectively increasing your sample size for quite a lot (from 32 to 2455). Smaller p-values are expected when you have a larger sample size and the null hypothesis is true.I would suggest to add something about this to the manuscript.

---

## Round 0.2 · Major Revisions

I am forwarding the reviewer comments on the previous submission, please address them line by line in your rebuttal. In your edits, make sure to add 95% CI or standard errors for all the proportions in tables 1 to 3. I realize that scatter plots generated by software programs commonly include regression lines but the issue raised by the reviewer may be valid if the software you used generated these lines based on parametric regression models.

While the regression line is meant to depict the trend lines, if the model used was parametric then its underlying assumption that all the regression points are independent is violated (because the data points are clustered by regions). Please check your software help files and confirm this.

If it is, then avoid this issue by considered replacing the traditional regression lines with locally weighted linear regression (LOESS) or kernel weighted local polynomial regression. These are non-parametric and hence the dependence assumption isn't violated.

·

Basic reporting

I found some gramatical errors in lines:
31 - Should be “to collect…”, instead of 'to collected..."
451 - Did you meant “viral diseases” instead “viral deceases”?

For some sentences I understand what you are trying to say, but I would suggest to try to improve it to ensure that the audience clearly understand, including sentences from lines:
123-124 “There are no standardized medical therapeutics or vaccines against COVID-19 are just coming out and there is probably no pre-existing immunity in the population”, if you meant to mention about the newly developed vaccines or the treatments, I think you need to paraphrase it better.
154 - I suggest to add the last part as a third objective, i.e. “c) investigate the correlation between the pandemic…”
284 - To avoid any confusion, I suggest to change ‘crude mortality rate’ for ’cause specific mortality’. Crude mortality rate could be interpreted of deaths by any cause.
454 - "The local (municipality) scale that consisted of 2455 data the only 32 data of federal scale showed typically “smaller p-values” decreasing the uncertainty of the analysis" This sentence is a bit unambiguous please be more specific regarding what you mean with "typical smaller p-values" part.

The raw data looks great

Experimental design

For the regression, I suggest to also provide more details such as the p-value and/or confidence intervals for the coefficients, which would provide a better idea of the uncertainty of your estimations.

Last time I suggested you to use disease specific mortality, which I really meant to say disease fatality (deaths/cases) I apologize for my mistake, but I want to mention it because I think it should be very quick to calculate and could provide you interesting information for your tables and thematic maps.

Validity of the findings

My main concern regarding the regression model used, is that you didn’t explored or even mentioned anything about spatial autocorrelation. Remember that one of the assumptions of regression is independence, which is violated when you have autocorrelated samples.
So maybe consider expanding on the discussion about spatial autocorrelation mentioning it as a limitation or including a random effect for the municipal level model to account for this.

---

## Round 0.3 · accepted · Accept

Thank you for addressing the reviewer comments, I find your manuscript acceptable for publication.

Reviewer 1 ·

Basic reporting

no comment

Experimental design

no comment

Validity of the findings

no comment

Additional comments

no comment

·

Basic reporting

The grammar and structure of the text has improved, well done

Experimental design

The authors made several changes after the suggestions we did, great work!

Validity of the findings

Everything looks ok.

Additional comments

Thanks for addressing the comments, I am satisfied with the improvements you have made to the manuscript.